# Taming the Chaos in Neural Network Time Series Predictions

**DOI:** 10.3390/e23111424

**Published:** 2021-10-28

**Authors:** Sebastian Raubitzek, Thomas Neubauer

**Affiliations:** Information and Software Engineering Group, Institute of Information Systems Engineering, Faculty of Informatics, TU Wien, Favoritenstrasse 9-11/194, 1040 Vienna, Austria; thomas.neubauer@tuwien.ac.at

**Keywords:** Hurst exponent, chaos, Lyapunov exponents, neural networks, time series prediction, deep learning, machine learning, LSTM, R/S analysis

## Abstract

Machine learning methods, such as Long Short-Term Memory (LSTM) neural networks can predict real-life time series data. Here, we present a new approach to predict time series data combining interpolation techniques, randomly parameterized LSTM neural networks and measures of signal complexity, which we will refer to as complexity measures throughout this research. First, we interpolate the time series data under study. Next, we predict the time series data using an ensemble of randomly parameterized LSTM neural networks. Finally, we filter the ensemble prediction based on the original data complexity to improve the predictability, i.e., we keep only predictions with a complexity close to that of the training data. We test the proposed approach on five different univariate time series data. We use linear and fractal interpolation to increase the amount of data. We tested five different complexity measures for the ensemble filters for time series data, i.e., the Hurst exponent, Shannon’s entropy, Fisher’s information, SVD entropy, and the spectrum of Lyapunov exponents. Our results show that the interpolated predictions consistently outperformed the non-interpolated ones. The best ensemble predictions always beat a baseline prediction based on a neural network with only a single hidden LSTM, gated recurrent unit (GRU) or simple recurrent neural network (RNN) layer. The complexity filters can reduce the error of a random ensemble prediction by a factor of 10. Further, because we use randomly parameterized neural networks, no hyperparameter tuning is required. We prove this method useful for real-time time series prediction because the optimization of hyperparameters, which is usually very costly and time-intensive, can be circumvented with the presented approach.

## 1. Introduction

Machine learning and neural networks are today’s state of the art when it comes to predicting time series data. Applications feature various research areas and tasks, such as future population estimates, predicting epileptic seizures [1], or estimating future stock market prices.

All machine learning approaches depend on the quality and quantity of the available data, i.e., their complexity or randomness and the actual amount of data, and the algorithm’s right parameterization. The three main reasons for machine learning approaches to perform poorly are:An insufficient amount of data, i.e., the data are not fine-grained or long enough;Random data, with its blueprint, the Brownian motion (cf. [2]);Bad parameterization of the algorithm.

A way to enrich a dataset and increase its fine-grainededness is to use interpolation. In the simplest case, this is done using a linear interpolation [3]. Complex real-life data are generally considered non-linear and originate from systems where the interactions and corresponding agents are not known in detail. Therefore, linear interpolation approaches do not depict the complexity and non-linearity of real-life systems. One approach to consider the complexity of the data under study is fractal interpolation [4]. The challenge is to find the right interpolation technique for the task at hand, i.e., what interpolation technique is best suited for improving machine learning time series predictions?

There are many ways to measure the complexity or inherent information of time series data. This article will refer to these measures of signal complexity as *complexity measures* since they measure how complex the data under study are. Given that, the question is how can the complexity of the data under study be taken into account to improve predictions?

Finding the right set of hyperparameters for any given machine learning algorithm, especially for neural networks, can be a tedious and time-consuming task. In addition, there is usually more than one set of hyperparameters with high performance for a given dataset. Here, the hypothesis is that differently parameterized algorithms, again especially neural networks, capture different aspects of the data. However, is there a way to take into account different aspects of the data to reduce the effort of finding the right set of parameters?

We present an approach that addresses those three issues. The overall hypothesis here is that neural network time series predictions can be improved by considering the complexity of the data under study. First, we interpolate the data under study using a fractal interpolation adapted to the data complexity. Second, we use randomly parameterized long short-term memory (LSTM [5]) neural networks to make ensemble predictions. Third, we use complexity properties of the data for filtering the ensemble predictions to improve their accuracy. Using the presented ideas, we can reduce the error of a random long short term memory (LSTM [5]) neural network ensemble prediction on average by a factor of 10 by filtering the results depending on their complexity. Further, we set a baseline by using an LSTM, a GRU, [6], and a recurrent neural network, [7], with only one hidden layer to make predictions of the time series data under study. The best ensemble predictions always outperformed these baseline predictions. The discussed filter methods can be applied to any ensemble predictions and therefore be used to improve existing ensemble approaches.

We show that the fractal interpolation approach, which considers the complexity of the data under study, is the preferred method to improve neural network time series predictions. Therefore, one should consider the discussed interpolation techniques when facing an insufficient amount of data.

Given that we used a randomly parameterized neural network implementation, this approach is a more generalized one and can be applied to any univariate time series dataset without further optimization, i.e., we can circumvent the optimization procedure. In addition, since the ensemble consists of several independent neural networks, this approach can, in principle, be parallelized infinitely to speed up the calculations.

We further show how different filters perform for different datasets and give recommendations on which filters to employ to improve ensemble predictions.

The remainder of the paper is organized as follows. Section 2 gives an overview on related research. In Section 3, we describe the used methodology and discuss how the different techniques fit together. Section 4 presents all used datasets and Section 5 the corresponding interpolation techniques. In Section 6, we discuss the used complexity measures. Section 7 shows the neural network implementation and the ensemble scheme. The error metrics are discussed in Section 8. Section 9 describes the used ensemble (-complexity) filters. In Section 10, we present LSTM, GRU and recurrent neural network predictions as a baseline to compare our ensemble predictions to. Section 11 discusses the results. We further conclude our findings in Section 12. In addition, we added an Appendix to collect peripheral plots and data to keep the main text focused.

## 2. Related Work

Machine learning and neural networks are the state of the art when it comes to time series predictions. Tools, such as support vector regression, are used for a variety of applications, as shown in [8]. Here, the often studied problems are the forecasting of financial markets [9,10] and predicting electrical load forecasting [11]. Other applications are to predict solar radiation [12] with, e.g., artificial neural networks (ANNs). In general, we expect neural networks to perform best for various types of data [13]. Popular types of neural networks used for predictions are recurrent neural networks (RNNs), as described in [14,15], or long short term memory neural networks (LSTM) [5]. LSTM neural networks are a category of RNNs [16] and are a class of artificial neural networks. RNNs are capable of using recurrent connections, i.e., a directed graph along a temporal sequence. Though, in theory, this is supposed to be very effective when analyzing time series, RNNs suffer from vanishing or exploding gradients when trained with Gradient Descent. LSTMs are designed to solve this issue and also to learn inherent long-term dependencies, and as such, are capable of predicting time series data. For these reasons, we chose to use LSTM neural networks in this study.

A feature specific for LSTMs is the so-called *memory block*, which enhances the LSTM’s capability to learn and model long-term dependencies. This memory block is a subnet of the actual neural network, which is recurrently connected. Two functional modules are part of this subnet, the memory cell and corresponding gates. The memory cell serves to remember the neural network’s temporal state, and the corresponding gates control the information flow and are multiplicative units. Three types of gates are used, input gates, output gates, and forget gates. The input gates, just like the name says, control how much information goes in the cell, and the output gates control the returned information, i.e., fuel the output activation. The forget gates, on the other hand, are responsible for containing information in the cell. All these mechanics of the LSTM are to serve the task at hand as best as possible.

In addition, ensemble methods, i.e., combinations of different neural networks or machine learning algorithms (or both), prove useful for predicting time series data, as done in [17] to forecast exchange rates. Here, the traditional approach is to *keep-the-best* (KTB). State-of-the-art is to use different neural network architectures to capture different aspects of the data, e.g., [18], where an ensemble of LSTM neural networks (in conjunction with other non-linear methods) is used to forecast wind speeds. Further, in [19], ensemble predictions could be improved when adding noise to the data under study, somehow similar to the noise added using fractal interpolation in this research.

When it comes to interpolation techniques to improve machine learning applications, one is tempted to use a linear interpolation as done in [3]. However, just as the name says, a linear interpolation is only a linear fit between some data points. One approach to consider the complexity of the data under study is fractal interpolation [4]. Traditional interpolation methods are based on elementary functions, such as polynomials. Fractal interpolation, in contrast, is based on iterated function systems. Iterated function systems can generate fractal and multi-fractal structures, therefore preserving the inherent complexity of the original data.

Measuring the complexity/information/randomness (i.e., non-linear properties) of given time series data can be done in many ways. One example is the Hurst exponent [20], which was found to be a measure for long-term memory of time series data. Other information measures use all sorts of entropy measures that can be applied to time series data, such as Shannon’s entropy, as used in [21] to analyze astronomical time series data. See Section 6 for all employed complexity measures.

There are very few approaches combining complexity measures and machine learning yet. In [22], the local Hölder exponent is used as an additional complexity feature to the time series to improve predictions. In [3], the Hurst exponent, Rényi entropy and Shannon’s entropy are employed to improve forecasts of financial markets and cryptocurrencies. In [23], ideas from fuzzy logic and fractal theory improve neural networks time series predictions. Further, in [24], a fractal interpolation taking into account the complexity of the data under study is used to improve LSTM time series predictions. As such, this research is based on the findings of [24].

## 3. Methodology

We apply a fractal interpolation method and a linear interpolation method to five datasets to increase the data fine-grainededness. The fractal interpolation was tailored to match the original data complexity using the Hurst exponent. Afterward, random LSTM neural networks are trained and used to make predictions, resulting in 500 random predictions for each dataset. These random predictions are then filtered using Lyapunov exponents, Fisher information and the Hurst exponent, and two entropy measures to reduce the number of random predictions. Here, the hypothesis is that the predicted data have to have the same complexity properties as the original dataset. Therefore, good predictions can be differentiated from bad ones by their complexity properties.

As far as the authors know, a combination of fractal interpolation, complexity measures as filters, and random ensemble predictions in this way has not been presented yet.

We developed a pipeline connecting interpolation techniques, neural networks, ensemble predictions, and filters based on complexity measures for this research. The pipeline is depicted in Figure 1.

First, we generated several different fractal-interpolated and linear-interpolated time series data, differing in the number of interpolation points (the number of new data points between two original data points), i.e., {1, 3, 5, 7, 9, 11, 13, 15, 17} and split them into a training dataset and a validation dataset. (Initially, we tested if it is necessary to split the data first and interpolate them later to prevent information to leak from the train data to the test data. However, that did not make any difference in the predictions, though it made the whole pipeline easier to handle. This information leak is also suppressed as the interpolation is done sequentially, i.e., for separated subintervals.) Next, we generated 500 randomly parameterized long short-term memory (LSTM) neural networks and trained them with the training dataset. Then, each of these neural networks produces a prediction to be compared with the validation dataset. Next, we filter these 500 predictions based on their complexity, i.e., we keep only those predictions with a complexity (e.g., a Hurst exponent) close to that of the training dataset. The remaining predictions are then averaged to produce an ensemble prediction.

## 4. Datasets

For this research, we tested five different datasets. All of them are real-life datasets, and some are widely used for time series analysis tutorials. All of them are contributed to [25] and are part of the *Time Series Data Library*. They differ in their number of data points and their complexity (see Section 6).
**Monthly international airline passengers:** January 1949 to December 1960, 144 data points, given in units of 1000. Source: Time Series Data Library, [25];**Monthly car sales in Quebec:** January 1960 to December 1968, 108 data points. Source: Time Series Data Library [25];**Monthly mean air temperature in Nottingham Castle:** January 1920 to December 1939, given in degrees Fahrenheit, 240 data points. Source: Time Series Data Library [25];**Perrin Freres monthly champagne sales:** January 1964 to September 1972, 105 data points. Source: Time Series Data Library [25];**CFE specialty monthly writing paper sales.** This dataset spans 12 years and 3 months, 147 data points. Source: Time Series Data Library [25].

## 5. Applied Interpolation Techniques

We apply two different interpolation techniques to generate more data points for the discussed datasets: Fractal interpolation and linear interpolation. We used the interpolation techniques to generate new interpolated datasets differing in the number of interpolation points, i.e., new data points between every two original data points. The interpolations were done for the following numbers $N_{I}=\left\{1,\;3,\;5,\;7,\;9,\;11,\;13,\;15,\;17\right\}$.

### 5.1. Fractal Interpolation of Time Series Data

For the fractal interpolation, we employ a method developed in [24]. With the actual interpolation described in [4]. Therefore, we only give a summary of this method and refer to the sources for further reading.

In contrast to traditional interpolation approaches based on polynomials, fractal interpolation is based on iterated function systems. Iterated function systems are defined as a complete metric space *X* with a corresponding distance function *h* and a finite set of contractive mappings, $\left\{w_{n}\;:\;X\to X\;\mathrm{for}\;n\;=\;1,\;2,\;\ldots ,\;N\right\}$ [26]. For further reading on iterated function systems, we refer to [27].

A time series is given as a set of data points as $\{\left(u_{m},\;v_{m}\right)\;\in \;R^{2}\}:\;m\;=\;0,\;1,\;\ldots ,M$. The interpolation is then applied to a subset of those data points, i.e., the interpolation points $\left\{\left(x_{i},\;y_{i}\right)\;\in \;R^{2}:\;i=\;0,\;1,\;\ldots ,N\right\}$. Both sets are linearly ordered with respect to their abscissa, i.e.: $u_{0}\;<\;u_{1}\;<\;\ldots \;<\;u_{M}$ and $x_{0}\;<\;x_{1}\;<\;\ldots \;<\;x_{M}$. The data points are then partitioned into intervals by the interpolation points. For our implementation, the interpolation intervals are chosen to be equidistant. The more interpolation points are used, the better the interpolation fits the original data. However, more interpolation points result in a smaller compression ratio since more information is needed to describe the interpolation function. This ratio, respectively, is the ratio of the information of the original data and the information of the interpolated data.

An iterated function system is given as $\left\{R^{2};\;w_{n},\;n\;=\;1,\;2,\;\ldots ,\;N\right\}$ with the corresponding affine transformations
(1)$$w_{n}\left[\begin{array}{c} x\\ y \end{array}\right]\;=\;\left[\begin{array}{cc} a_{n} & 0\\ c_{n} & s_{n} \end{array}\right]\;\left[\begin{array}{c} x\\ y \end{array}\right]+\left[\begin{array}{c} d_{n}\\ e_{n} \end{array}\right]\;,$$
which satisfy
(2)$$w_{n}\left[\begin{array}{c} x_{0}\\ y_{0} \end{array}\right]\;=\;\left[\begin{array}{c} x_{n-1}\\ y_{n-1} \end{array}\right]\;\mathrm{and}\;w_{n}\left[\begin{array}{c} x_{N}\\ y_{N} \end{array}\right]\;=\;\left[\begin{array}{c} x_{n}\\ y_{n} \end{array}\right]\;,$$
for every n = 1, 2, …, N. Solving these equations yields
(3)$$\begin{array}{cc} & a_{n}\;=\;\frac{x_{n}-x_{n-1}}{x_{N}-x_{0}}\;, \end{array}$$
(4)$$\begin{array}{cc} & \;\;d_{n}\;=\;\frac{x_{N}x_{n-1}-x_{0}x_{n}}{x_{N}-x_{0}}\;, \end{array}$$
(5)$$\begin{array}{cc} & \;\;\;\;\;c_{n}\;=\;\frac{y_{n}-y_{n-1}}{x_{N}-x_{0}}-s_{n}\frac{y_{N}-y_{0}}{x_{N}-x_{0}}\;, \end{array}$$
(6)$$\begin{array}{cc} & \;\;\;\;\;\;\;\;e_{n}\;=\;\frac{x_{N}y_{n-1}-x_{0}y_{n}}{x_{N}-x_{0}}-s_{n}\frac{x_{N}y_{0}-x_{0}y_{N}}{x_{N}-x_{0}}\;. \end{array}$$

The interpolation points determine the real numbers $a_{n},\;d_{n},\;c_{n},\;e_{n}$ and the *vertical scaling factor*
$s_{n}$ is a free parameter. $s_{n}$ is bounded by $\left|s_{n}\right|<1$ so that the IFS is hyperbolic with respect to an appropriate metric. Later on, $s_{n}$ is the parameter used to ensure the IFS fits the original data the way we want it.

### 5.2. Fractal Interpolation Applied

The following procedure, from [24], was applied to every time series to find a fractal interpolation that reproduces real-life complex time series data:Divide time series into *m* sub-sets of size *l*;For each sub-set *i*, calculate the corresponding Hurst exponent $H_{i}$;For each subset *i*, the following routine is performed k=500 times:(a)Use the fractal interpolation method from Section 5.1 with a random parameter $s_{n}$, where $s_{n}$ was set constant for the whole sub-set;(b)Calculate the Hurst exponent $H_{i,\mathrm{int},\mathrm{new}}$ for the interpolated time series;(c)If there was $H_{i,\mathrm{int},\mathrm{old}}$ set beforehand, compare it to $H_{i,\mathrm{int},\mathrm{new}}$. If $H_{i,\mathrm{int},\mathrm{new}}$ is closer to $H_{i}$, keep $s_{n}$, and the corresponding fractal interpolation and $H_{i,\mathrm{int},\mathrm{old}}$ is set to $H_{i,\mathrm{int},\mathrm{new}}$.

Remarks:The Hurst exponent was calculated using *R/S Analysis* [20];The number of iterations *k* was set to 500 for each dataset;No threshold was set for the Hurst exponent of the interpolated time series to match the one from the original time series, since, for some sub-intervals, several thresholds that have been tried could not be reached.

In Figure 2, the fractal interpolation is shown for the monthly international airline passengers dataset.

### 5.3. Linear Interpolation

The second method that was used to increase the number of data points is linear interpolation. Therefore, the original data $\{\left(u_{m},\;v_{m}\right)\;\in \;R^{2}\}:\;m\;=\;0,\;1,\;\ldots ,M$ is interpolated using a linear fit $y_{i}\;=\;a_{m}x_{i}+b_{m}$, to obtain the interpolation points $\left\{\left(x_{i},\;y_{i}\right)\;\in \;R^{2}:\;i=\;0,\;1,\;\ldots ,N\right\}$. This was done for each interval $\left[u_{m},u_{m+1}\right]$. The coefficients $a_{m}\;\mathrm{and}\;b_{m}$ are calculated using
(7)$$\begin{array}{c} a_{m}\;=\;\frac{v_{m+1}-v_{m}}{u_{m+1}-u_{m}}\;\mathrm{and}\;b_{m}\;=\;v_{m}-a_{m}u_{m}\;,\\ \forall \;m\;=\;0,\;1,\;\ldots ,M-1\;. \end{array}$$

## 6. Measuring the Complexity of the Data

First, we used the Hurst exponent to adjust the fractal interpolation (see Section 5.1) to be as close as possible to the original data in terms of its Hurst exponent.

Second, we compared the complexities, i.e., the complexity measures presented in this section, with the original complexity as a function of the interpolation points and the method used, i.e., fractal or linear interpolation.

Third, we used the complexity measures as a filter to improve the random ensembles’ accuracy which is discussed in Section 9.

The results presented in this section are not all complexity measures considered in this research. Before using the listed complexity/information/entropy measures, we performed an evaluation of additional complexity measures, including *detrended fluctuation analysis* (DFA) [28], some algorithms to calculate the *fractal dimension* [29] of a time series and the *generalized Hurst exponent* [30]. The complexity measures did not make it into the results because they performed poorly in the initial tests. We mention them here to be useful for future time series analysis approaches.

The *fractal dimension* of time series data was excluded because of its similarity to the *Hurst exponent* and *R/S analysis* [29].

In the following, we give an overview of the used complexity measures:

### 6.1. The Hurst Exponent (R/S Analysis)

The Hurst exponent is a measure of long term memory in time series data and it is calculated using *R/S Analysis* [20]. We only outline the main aspects of R/S analysis, for an in-depth treatment of the topic we refer to [20,31].

The rescaled range analysis (*R/S analysis*) is a technique to analyze long-run correlations in signals, and yields one parameter, the *Hurst exponent* “*H*”.

Given a signal $\left[x_{1},\;x_{2},\ldots ,\;x_{n}\right]$, the average over a period τ (a sub-interval of the signal, i.e., 1≤τ≤n), with a corresponding *k* as 1≤k≤n and elements *i* in this interval such that k≤i≤k+τ.
(8)$${\langle x\rangle }_{\tau ,k}\;=\;\frac{1}{\tau }\sum_{j=k}^{k+\tau }\;x_{j}\;.$$

The accumulated departure $\delta x\left(i,\;\tau ,\;k\right)$ over a period i∈1, 2,…, τ is:(9)$$\delta x\left(i,\;\tau ,\;k\right)\;=\;\sum_{j=k}^{i}\left(x_{j}-{\langle x\rangle }_{\tau ,k}\right)\;$$

The difference between maximal and minimal values of all $x_{i}$ in the interval $\left[k,\;k+\tau \right]$ is referred to as the range *R* of this interval τ:(10)$$\begin{array}{c} R\left(\tau ,\;k\right)\;=\;\max\left[\delta x\left(i,\tau ,k\right)\right]\;-\;\min\left[\delta x\left(i,\tau ,k\right)\right]\;,\\ \mathrm{satisfying}\;k\leq i\leq k+\tau \;. \end{array}$$

We further use the standard deviation for each sub-interval:(11)$$S\left(\tau ,\;k\right)\;=\;\sqrt{\frac{1}{\tau }\sum_{i=k}^{k+\tau }{\left[x_{i}-{\langle x\rangle }_{\tau ,k}\right]}^{2}}\;.$$

The next step is then to average the range and the standard deviation over all possible *k* as
(12)$$\begin{array}{c} R\left(\tau \right)\;=\;\frac{\sum_{k}R\left(\tau ,k\right)}{\mathrm{number}\mathrm{of}\mathrm{different}ks}\\ \;\mathrm{and}\\ S\left(\tau \right)\;=\;\frac{\sum_{k}S\left(\tau ,\;k\right)}{\mathrm{number}\;\mathrm{of}\;\mathrm{different}\;ks}\;, \end{array}$$
where 1≤k≤n and k≤i≤k+τ. The *Hurst exponent H* is then defined using the scaling properties as
(13)$$\frac{R\left(\tau \right)}{S\left(\tau \right)}\;\propto \;\tau^{H}\;.$$

The asymptotic behavior, [20] and [32], for any independent random process with finite variance is given as
(14)$$\frac{R\left(\tau \right)}{S\left(\tau \right)}\;=\;{\left(\frac{\pi }{2v}\tau \right)}^{\frac{1}{2}}\;,$$
where *v* is the equidistant interval between the data points of the signal $x_{i}$. This implies $H=\frac{1}{2}$, however, only for completely random processes. For real-life data, $H\neq \frac{1}{2}$, because real-life processes usually feature long-term correlations. The value of *H* is set to 0<H<1. A value H<0.5 indicates anti-persistency, meaning that low values follow on high values and vice versa, thus heavily fluctuating, but not totally random. Values close to 0 prove strong anti-persistency. Contrarily, values of H>0.5 indicate persistent behavior and strong persistence for values close to 1. It is also important to note that time series with H≠0.5 can theoretically be forecast [33].

For *R/S analysis* and to calculate the Hurst exponent, we used the algorithm provided by the python package nolds, https://pypi.org/project/nolds/, accessed on 20 October 2021 [34].

### 6.2. The Lyapunov Exponents’ Spectrum

The spectrum of Lyapunov exponents is a measure for a system’s predictability depending on initial conditions. When it comes to experimental time series data, as there are not different trajectories for different initial conditions available, we still interpret it as a measure for predictability [35]. As the calculation and the corresponding algorithm to calculate Lyapunov exponents of experimental data are too complex to be presented here, we refer to [35] for an in-depth discussion of the topic.

The employed algorithm from the python package nolds [34] yields the first 4 exponents of the spectrum. When referring to the largest Lyapunov exponent, we thereby mean the first one of the spectrum. In general, a positive Lyapunov exponent is a strong indicator for chaos [36]. Therefore, in most cases, it is sufficient to calculate the first Lyapunov exponent of the spectrum, hence the largest Lyapunov exponent. Systems possessing several positive Lyapunov exponents are referred to as hyperchaotic [37].

### 6.3. Fisher’s Information

Fisher’s information can be interpreted as the amount of information extracted from a set of measurements, i.e., the quality of the measurements [38]. Additionally, it can be interpreted as a measure of order or disorder of a system or phenomenon. It can also be used to investigate non-stationary and complex signals, hence ideally suited for this study.

The discrete version of Fisher’s information is suitable for univariate time series analysis, given as a signal $\left[x_{1},\;x_{2},\ldots ,\;x_{n}\right]$.

First, we construct embedding vectors as:(15)$$\vec{y}\left(i\right)\;=\;\left[x_{i},\;x_{i+\tau },\ldots ,\;x_{i+\left(d_{E}-1\right)*\tau }\right]\;,$$
with time delay τ and an embedding dimension $d_{E}$. The embedding space, as a matrix, then is:(16)$$Y\;=\;{\left[\vec{y}\left(1\right),\;\vec{y}\left(2\right),\ldots ,\;\vec{y}\left(N-\left(d_{E}-1\right)\tau \right)\right]}^{T}.$$

On this matrix, we perform a single value decomposition, [39], yielding *M* singular values $\sigma_{i}$ with the corresponding normalized singular values:(17)$${\bar{\sigma }}_{i}\;=\;\frac{\sigma_{i}}{\sum_{j=1}^{M}\sigma_{j}}\;.$$

Thus, we find Fisher’s Information as
(18)$$I_{\mathrm{Fisher}}\;=\;\sum_{i=1}^{M-1}\frac{{\left[{\bar{\sigma }}_{i+1}-{\bar{\sigma }}_{i}\right]}^{2}}{{\bar{\sigma }}_{i}}.$$

Here, the implementation from the python package neurokit, https://neurokit.readthedocs.io/en/latest/, accessed on 28 October 2021 [40] was used. This implementation requires two parameters, first the time delay, which was found using the calculation of the average mutual information from [41] and the embedding dimension which was set to 3 for all time series data, a possible alternative for detecting the embedding dimension would be to use the *False Nearest Neighbours* algorithm [42] to determine the embedding dimension. The reason for this is that Takens’ theorem [43] guarantees that a correct embedding space for systems consisting of *m* ordinary coupled differential equations is 2m+1 dimensional. For real-life data, however, we cannot employ a proper model of differential equations and, therefore, stick with 3. Further, one can find a discussion on why we chose 3 and the actual estimated embedding dimensions in Appendix F. In addition, since we define the interpolated and non-interpolated time series data to be of same origin, it makes sense to keep the embedding dimension $d_{E}$ constant for different interpolations but not the time delay τ.

### 6.4. SVD Entropy

SVD entropy (Single Value Decomposition) is an entropy measure based on the correlation matrix of a time series and a corresponding single value decomposition. It is known to be applicable to stock market data as a prediction criterion as done in [44,45].

SVD entropy is calculated by constructing an embedding space for a signal $\left[x_{1},\;x_{2},\ldots ,\;x_{n}\right]$ with delay vectors as [46]:(19)$$\vec{y}\left(i\right)\;=\;\left[x_{i},\;x_{i+\tau },\ldots ,\;x_{i+\left(d_{E}-1\right)*\tau }\right]\;,$$
with the corresponding time delay τ and the embedding dimension $d_{E}$. We construct the embedding space as the matrix:(20)$$Y\;=\;{\left[\vec{y}\left(1\right),\vec{y}\left(2\right),\ldots ,\vec{y}\left(N-\left(d_{E}-1\right)\tau \right)\right]}^{T}.$$

For this matrix, a single value decomposition, [39], is then performed to get *M* singular values $\sigma_{1},\ldots ,\sigma_{M}$ known as the singular spectrum. Further, we find the corresponding spectrum of normalized singular values as:(21)$${\bar{\sigma }}_{i}\;=\;\frac{\sigma_{i}}{\sum_{j=1}^{M}\sigma_{j}}.$$

Using the formula for Shannon’s entropy then yields SVD entropy as:(22)$$H_{\mathrm{SVD}}\;=\;\sum_{i=1}^{M}{\bar{\sigma }}_{i}\log_{2}{\bar{\sigma }}_{i}$$

We used the implementation from the python package neurokit [40]. For this algorithm, just as for the algorithm for Fisher’s information, we have to find two additional parameters, i.e., the embedding dimension and the time delay. Both were found the same way as above for Fisher’s Information.

### 6.5. Shannon’s Entropy

Given a signal $\left[x_{1},\;x_{2},\ldots ,\;x_{n}\right]$, we then find the probability to occur for each value denoted as $P\left(x_{1}\right),\ldots ,\;P\left(x_{n}\right)$, thus, we formulate Shannon’s entropy [47] as:(23)$$H_{\mathrm{Shannon}}\;=\;-\sum_{i=1}^{n}P\left(x_{i}\right)\log_{2}P\left(x_{i}\right)\;.$$
Giving units as bits, the base of the logarithm is set to 2. Applications include astronomy [21], where it is used to identify periodic variability, or in finance [48], as to measure the diversity of portfolios or to estimate risks. Shannon’s entropy is a measure for the uncertainty of a (random) process/signal.

### 6.6. Initial Complexity

We applied the five featured complexity measures to the original datasets. The results can be found in Table 1. We briefly discuss the complexities for each measure separately, as an in-depth discussion with regards to their predictability can be found in Section 11:**The Hurst exponent:** The most persistent dataset, with a Hurst exponent of 0.7988, is the dataset of monthly car sales in Quebec. According to [33], we expected that time series data with a very high Hurst exponent can be predicted with higher accuracy than ones with a value close to 0.5, as it is considered more random. The datasets under study are three persistent ones, i.e., with a Hurst exponent larger than 0.5. Contrary to that, we obtained two anti-persistent ones with a Hurst exponent below 0.5;**The largest Lyapunov exponent:** All largest Lyapunov exponents of all time series data under study are positive, just as we would expect from chaotic or complex real-life data. The dataset with the highest value is the monthly car sales data in the Quebec dataset. As the Lyapunov exponents of experimental time series data serve as a measure for predictability, we suggest this dataset, therefore, to be forecast with the least accuracy;**Fisher’s information:** The dataset with the highest value of Fisher’s information is the monthly international airline passengers dataset. The lowest value can be found for the Perrin Freres monthly champagne sales dataset. [38]: It is expected that Fisher’s information behaves contrary to entropy measures since it is a measure for order/quality, which we observed only for SVD entropy. This means that the Fisher’s information value for the monthly international airline passengers dataset is the highest, and the corresponding value for the SVD entropy is the lowest among all datasets. The reason for this may be that, just like Fisher’s information, SVD entropy is based on a Single Value Decomposition [39]. Considering Shannon’s entropy, which is also an entropy measure but not one based on Single Value decomposition, its behavior differs;**SVD entropy:** The largest value of SVD entropy is possessed by the Perrin Freres monthly champagne sales dataset, which has, just as expected, the lowest value of Fisher’s information. We expect, as it has a very high SVD entropy value, the Perrin Freres champagne sales dataset not to be predicted with high accuracy;**Shannon’s entropy:** Shannon’s entropy is based on the frequency of occurrence of a specific value. As we deal with non-integer-valued complex datasets, we expect Shannon’s entropy not to be of much use. Shannon’s entropy’s highest value is found for the monthly mean temperature in Nottingham Castle dataset. Since reoccurring temperature distributions possess a higher regularity than, e.g., airline passengers, this explains the corresponding value.

### 6.7. Complexity Plots

We compared the complexities of all time series data under study, i.e., for different interpolation techniques and different numbers of interpolation points.

The results are shown in Figure 3, Figure 4 and Figure 5 for the monthly international airline passengers dataset, the plots for all the other time series data can be found in Appendix A. Note that, in each of the plots, the blue line, i.e., the complexity of the non-interpolated time series is not a plot depending on the number of interpolation points, but a constant, since there is only one dataset, the one with zero interpolation points, and plotted as a line for reference. In addition, those original complexities are contained in Table 1.

For this study, we observe the following behavior:The Hurst exponent of the fractal and the linear interpolated datasets behave very similar to each other for the monthly international airline passengers dataset, see Figure 3. We observe similar behavior for the other datasets as well, see Appendix A. Though the Hurst exponent is initially lower for the fractal-interpolated data for some datasets, the Hurst exponent does not differ significantly between fractal and linear interpolated time series data. In addition, adding more interpolation points increases the Hurst exponent and makes the datasets more persistent;The Largest Lyapunov exponents of the fractal-interpolated data are much closer to the original data than the ones for the linear-interpolated data; see Figure 4. We observe the same behavior for all datasets; see Appendix A;Fisher’s information for the fractal-interpolated dataset is closer to that of the original dataset (see Figure 3). We observe the same behavior for all datasets, as can be seen in Appendix A;Just as expected, SVD entropy behaves contrary to Fisher’s information. In addition, the SVD entropy of the fractal interpolated time series is closer to that of the non-interpolated time series; see Figure 5. The same behavior and, specifically, the behavior contrary to that of Fisher’s information can be observed for all datasets under study; see Appendix A;Shannon’s entropy increases. This can be explained as follows: As more data points are added, the probability of hitting the same value increases. However, this is just what Shannon’s entropy measures. For small numbers of interpolation points, Shannon’s entropy of the fractal interpolated time series data is closer to the original complexity than the linear interpolated time series data. For large numbers of interpolation points, Shannon’s entropy performs very similarly, not to say overlaps, for the fractal- and linear-interpolated time series data. This behavior can be observed for all datasets, see Figure 4 and Appendix A.

Summing up our findings of the complexity analysis above, we find that:The fractal interpolation captures the original data complexity better, compared to the linear interpolation. We observe a significant difference in their behavior when studying SVD entropy, Fisher’s information, and the largest Lyapunov exponent. This is especially true for the largest Lyapunov exponent, where the behavior completely differs. The largest Lyapunov exponent of the fractal interpolated time series data stays mostly constant or behaves linearly. The largest Lyapunov exponent of the linear-interpolated data behaves approximately like a sigmoid function, and for some datasets even decreases again for large numbers of interpolation points.Both Shannon’s entropy and the Hurst exponent seem not suitable for differentiating between fractal- and linear-interpolated time series data.

## 7. LSTM Ensemble Predictions

For predicting all time series data, we employed random ensembles of different long short term memory (LSTM) [5] neural networks.

Our approach is to not optimize the neural networks but to generate many of them, in our case 500, and use the averaged results to obtain the final prediction.

For all neural network tasks, we used an existing keras 2.3.1 implementation.

### 7.1. Data Preprocessing

Two basic concepts of data preprocessing were applied to all datasets before the ensemble predictions.

First, the data X(t) defined at discrete time intervals *v*, thus t = v, 2v, 3, … kv, were scaled so that $X\left(t\right)\;\in \;\left[0,\;1\right],\;\forall t$. This was done for all datasets. Second, the data were made stationary by detrending them using a linear fit.

All datasets were split so that the first 70% were used as a training dataset and the remaining 30% to validate the results.

### 7.2. Random Ensemble Architecture

As previously mentioned, we used a random ensemble of LSTM neural networks. Each neural network was generated at random and consists of a minimum of 1 LSTM layer and 1 Dense layer and a maximum of 5 LSTM layers and 1 Dense layer.

Further, for all activation functions (and the recurrent activation function) of the LSTM layers, hard_sigmoid was used and relu for the Dense layer. The reason for this is that, at first, relu for all layers was used and we sometimes experienced very large results that corrupted the whole ensemble. Since hard_sigmoid is bound by [0, 1] changing the activation function to hard_sigmoid solved this problem. Here, the authors’ opinion is that the shown results can be improved by an activation function, specifically targeting the problems of random ensembles.

Overall, no regularizers, constraints or Drop out criteria have been used for the LSTM and Dense layers.

For the initialization, we used glorot_uniform for all LSTM layers, orthogonal as the recurrent initializer and glorot_uniform for the Dense layer.

For the LSTM layer, we also used use_bias=True, with bias_initializer="zeros" and no constraint or regularizer.

The optimizer was set to rmsprop and, for the loss, we used mean_squared_error. The output layer always returned only one result, i.e., the next time step. Further, we randomly varied many parameters for the neural network:**LSTM layers**: min 1, max 5;**Size of the input layer**: min 1, max depending on the size of the data, i.e., length of the training data −1;**Epochs**: min 1, max 30;**Neurons**: For both LSTM and Dense layers, min 1, max 30;**Batchsize**: The batchsize was chosen randomly from {1, 8, 32, 64, 128}.

For each prediction, 500 of these networks were created, trained, and generated a prediction for the unknown data.

## 8. Error Analysis

For each ensemble prediction, i.e., consisting of several different predictions labeled with *i* for each time step *t*, we calculated the mean and the standard deviation as
(24)$$\hat{X}\left(t\right)=\frac{1}{N_{p}}\sum_{i=1}^{N_{p}}{\hat{X}}_{i}\left(t\right)\;,\;\sigma \left(t\right)=\sqrt{\frac{1}{N_{p}}\sum_{i=1}^{N_{p}}{\left({\hat{X}}_{i}\left(t\right)-\hat{X}\left(t\right)\right)}^{2}}\;,$$
where ${\hat{X}}_{i}\left(t\right)$ is a single observation, $\hat{X}\left(t\right)$ is the averaged observation for a single time step, $\sigma \left(t\right)$ is the corresponding standard deviation and $N_{p}$ is the number of different predictions for each time step.

Next, to compare it to the validation dataset, we calculated the root-mean-square error (RMSE) as
(25)$$E_{RMSE}\;=\;\left(\frac{1}{N_{t}}\sum_{t=1}^{N_{t}}{\left[\hat{X}\left(t\right)\;-\;X\left(t\right)\right]}^{2}\right)\;,$$
where $X\left(t\right)$ are the data points of the validation dataset and $N_{t}$ is the number of validation data points. Using error propagation, the corresponding error of the root-mean-square error was calculated as
(26)$$\Delta E_{RMSE}=\sqrt{{\left(\frac{\partial E_{RMSE}}{\partial \hat{X}\left(1\right)}\right)}^{2}\sigma^{2}\left(1\right)+{\left(\frac{\partial E_{RMSE}}{\partial \hat{X}\left(2\right)}\right)}^{2}\sigma^{2}\left(2\right)+\;\ldots }\;,$$
thus yielding:(27)$$\Delta E_{RMSE}\;=\;\sqrt{\frac{\sum_{t=1}^{N_{t}}{\left[\hat{X}\left(t\right)\;-\;X\left(t\right)\right]}^{2}\;\sigma^{2}\left(t\right)}{N_{t}*\sum_{t=1}^{N_{t}}{\left[\hat{X}\left(t\right)\;-\;X\left(t\right)\right]}^{2}}}\;.$$

## 9. Complexity Filters

For each time series data, we produced statistics consisting of 500 random predictions for the non-interpolated data, for the fractal-interpolated data, and the linear-interpolated data. Averaging these predictions yields the ensemble results which can be found in Table A5 and Table A6. Since these averaged predictions are tainted with very bad predictions (because, after all, these are non-optimized, randomly parameterized neural networks), we propose that good predictions need to have a similar complexity as the original (fractal or linear-interpolated, respectively) time series. Therefore, we employed several filters to keep those predictions with complexities close to the original (fractal or linear-interpolated, respectively) time series. Therefore, five different filters were applied to all ensemble predictions so that only 1% of all predictions remained, i.e., 5 of 500 predictions. For an in-depth discussion of the employed complexity measures, see Section 6.
**Hurst exponent filter:** The Hurst exponent of each prediction of the ensemble was calculated and compared to the complexity of the corresponding training dataset: (The complexity of the fractal interpolated training data was compared with the complexity of the fractal-interpolated predictions, the same is true with linear-interpolated prediction and training datasets and non-interpolated training and test datasets.) (Note that this is crucial for real-time predictions as there is no validation dataset.)
(28)$$\Delta H=\left|\hat{H}-H\right|\;,$$
where $\hat{H}$ is the Hurst exponent of a prediction and *H* is the Hurst exponent of the training dataset. Afterward, only the ensemble predictions with Hurst exponents similar to the training dataset, i.e., with low ΔH are kept, and the others are discarded;**Lyapunov exponents filter:** The first 4 Lyapunov exponents of each prediction of the ensemble were calculated and compared with those of the training dataset.
(29)$$\Delta L=\sum_{i=1}^{4}\left|{\hat{L}}_{i}-L_{i}\right|\;$$
where ${\hat{L}}_{i}$ is the *i*th Lyapunov exponent of a prediction and $L_{i}$ is the *i*th Lyapunov exponent of the training dataset. Thus, only the predictions with the lowest ΔL are kept, and the others are discarded;**Fisher’s information filter:** Fisher’s information of each prediction of the ensemble was calculated and compared to Fisher’s information of the training dataset:
(30)$$\Delta I_{\mathrm{Fisher}}=\left|{\hat{I}}_{\mathrm{Fisher}}-I_{\mathrm{Fisher}}\right|\;,$$
where ${\hat{I}}_{\mathrm{Fisher}}$ is Fisher’s information of a prediction and $I_{\mathrm{Fisher}}$ is Fisher’s information of the training dataset. Afterward, only the predictions of the ensemble with Hurst exponents similar to the training dataset, i.e., with low $\Delta I_{\mathrm{Fisher}}$ are kept and the others are discarded;**SVD entropy filter:** The SVD entropy of each prediction of the ensemble was calculated and compared to the SVD entropy of the training dataset:
(31)$$\Delta H_{\mathrm{SVD}}=\left|{\hat{H}}_{\mathrm{SVD}}-H_{\mathrm{SVD}}\right|\;,$$
where ${\hat{H}}_{\mathrm{SVD}}$ is the SVD entropy of a prediction and $H_{\mathrm{SVD}}$ is the SVD entropy of the training dataset. Afterward, only the predictions of the ensemble with a SVD entropy similar to the training dataset, i.e., with low $\Delta H_{\mathrm{SVD}}$, are kept and the others are discarded;**Shannon’s entropy filter:** Shannon’s entropy of each prediction of the ensemble was calculated and compared to the Shannon’s entropy of the training dataset:
(32)$$\Delta H_{\mathrm{Shannon}}=\left|{\hat{H}}_{\mathrm{Shannon}}-H_{\mathrm{Shannon}}\right|\;,$$
where ${\hat{H}}_{\mathrm{Shannon}}$ is Shannon’s entropy of a prediction and $H_{\mathrm{Shannon}}$ is Shannon’s entropy of the training dataset. Afterward, only the predictions of the ensemble with Shannon’s entropy similar to the training dataset, i.e., with low $\Delta H_{\mathrm{Shannon}}$, are kept and the others are discarded.

In addition, all mentioned filters were applied in combination with each other, e.g., first, the Hurst exponent filter and afterward the Fisher’s information filter, to yield a remaining 1% of the ensemble. Therefore, the first filter reduces the whole ensemble to only 10%, i.e., 50 predictions, and the second filter reduces the remaining predictions to 10%, i.e., five predictions, thus 1%.

Figure 6 depicts the idea of the complexity filters. The left image shows the whole ensemble without any filtering, and the right side shows the filtered ensemble prediction. In this particular case, a filter combining SVD entropy and Lyapunov exponents was used to improve the ensemble predictions.

## 10. Baseline Predictions

For a baseline comparison, we use three types of recurrent neural networks. First, an LSTM neural network, second, a gated recurrent unit neural network (GRU) and, third, a simple recurrent neural network (RNN). All three employed types of neurons are reasonable tools for predicting time series data; the interested reader is referred to [6] for GRU and to [7] for simple RNN architectures for predicting time series data. We used a neural network with 1 hidden layer and 30 neurons, i.e., the RNN-neurons: LSTM, GRU, RNN. We used 20 input nodes consisting of consecutive time steps. The neural network was trained with a total of 50 epochs for each dataset. A batch size of 2 was used and verbose was set to 2 as well.

For the activation function (and the recurrent activation function) of the LSTM, the GRU and the SimpleRNN layer hard_sigmoid was used and relu for the Dense layer.

Overall, no regularizers, constraints or Drop out criteria have been used for the LSTM and Dense layers.

For the initialization, we used glorot_uniform for the LSTM layer, orthogonal as the recurrent initializer and glorot_uniform for the Dense layer.

For the LSTM layer, we also used use_bias=True, with bias_initializer="zeros" and no constraint or regularizer.

The optimizer was set to rmsprop and, for the loss, we used mean_squared_error. The output layer always returned only one result, i.e., the next time step.

These baseline predictions provide a reasonable guess for the accuracy of a LSTM, GRU or RNN prediction of the time series data under study.

All plots for the baseline predictions can be found in Appendix D, and here, we only give the accuracies for the test fit, the train fit and the single step-by-step prediction. These accuracies are shown in Table 2, Table 3 and Table 4. The accuracies such as the one for the ensemble predictions were calculated for linear-detrended and normalized (in the interval $\left[0,\;1\right]$) data.

## 11. Results and Discussion

We linear- and fractal-interpolated five different time series data. Afterward, we did a random ensemble prediction for each, consisting of 500 different predictions for each interpolation technique (and non-interpolated time series data). The results of these random ensembles can be found in Appendix E in Table A5 and Table A6. We further filtered these predictions using complexity filters (see Section 9) to finally reduce the number of ensemble predictions from 500 to 5, i.e., to 1%. The best five results for all time series data and each interpolation technique, regarding the RMSE and the corresponding error (see Section 8) are shown in Table 5 for the monthly international airline passengers dataset. Table A1, Table A2, Table A3 and Table A4, which feature the results for all other datasets, can be found in Appendix B. The corresponding plots for the three best predictions of each time series data can be found in Appendix C. We highlighted the overall best three results as bold entries.

The results show that the interpolated approaches always outperformed the non-interpolated ones when it comes to the lowest RMSEs. Further, the ensemble predictions could significantly be improved using a combination of interpolation techniques and complexity filters.

### 11.1. Interpolation Techniques

Regarding the different interpolation techniques of the overall three best results for all time series data, i.e., a total of 15 predictions, we find 9 fractal-interpolated predictions and 6 linear-interpolated predictions. Though the linear-interpolated results outperformed the fractal-interpolated ones in some cases, we conclude that fractal interpolation is a better way to improve LSTM neural network time series predictions. The reason for this is:

Taking into account the results shown in Figure 7 and Table 5, though the RMSE of the linear-interpolated result is lower (best result, lowest RMSE) than that of the second and third best ones (the fractal-interpolated results), the corresponding error of the RMSE is higher. Taking a closer look at the different ensemble predictions of Figure 7, we can see that the quality of the single predictions for the linear interpolated case is lower in terms of how close the actual curve data are to the different ensemble predictions. Therefore, the authors guess that this advantage of the linear-interpolated results vanishes when the statistic, i.e., the number of different ensemble predictions, increases. This behavior can be found for the monthly international airline passenger dataset, the monthly car sales in Quebec dataset, and the CFE specialty monthly writing paper sales dataset.

### 11.2. Complexity Filters

Of these 75 best results for all interpolation techniques and different data, only 13 are single filtered predictions. A significant 62 are double-filtered predictions (i.e., two different complexity filters were applied). Not a single unfiltered prediction made it into the top 75 results. We, therefore, suggest always using two different complexity filters for filtering ensembles.

When it comes to the specific filters used, we cannot find a pattern within the 15 best results, as only the combinations SVD entropy & Hurst exponent and Lyapunov exponents & Hurst exponent occur more than once, i.e., each occurred only two times. Examining the 75 best results, though, we get a different picture. Here, we find 7 occurrences of the combination Shannon’s entropy & Fisher’s information followed by 6 occurrences of Shannon’s entropy & SVD entropy. Further, taking into account that SVD entropy and Fisher’s information behave similarly (as both are based on SVD, see Section 6), we find that 57 of the best 75 results contain at least one SVD-based, i.e., Single-Value-Decomposition-based, complexity measure. Therefore, we suggest using an SVD-based complexity measure in combination with the Hurst exponent Shannon’s entropy. The authors’ recommendation is that the best combination is SVD entropy & Hurst exponent.

### 11.3. Remarks and Summary

Summing up the results of this research, we draw the following conclusions:Random ensemble predictions can significantly be improved using fractal and linear interpolation techniques. The authors recommend using a fractal interpolation approach as the shown results feature a more stable behavior than those for the linear interpolation;Random ensemble predictions can significantly be improved using complexity filters to reduce the number of predictions in an ensemble. Taking into account the unfiltered and non-interpolated results shown in Table A5 and Table A6 and comparing them to the best results, shown in Table 5 and Table A1, Table A2, Table A3 and Table A4, we see that the RMSE was reduced by a factor of ≈10 on average;The best results of the random ensemble, i.e., the single step-by-step predictions always outperformed the baseline predictions, Table 2, Table 3 and Table 4, and Appendix D. Here, we note that the given baseline predictions are probably not the best results that can be achieved with an optimized LSTM neural network but are still reasonable results and serve as baseline to show the quality of the ensemble predictions;Though the unfiltered results (Table A5 and Table A6) suggest a trend and a minimum for the errors depending on the number of interpolation points, this trend vanishes when applying complexity filters. Therefore, we could not find a trend for the number of interpolation points for any interpolation technique and any complexity filters.

Though this research used univariate time series data for analysis, our approach can be extended to arbitrary dimensional multivariate time series data. We expect multivariate prediction approaches to benefit from this research greatly. For multivariate time series, different features may have different complexity properties. Thus, one may employ tools such as effective transfer entropy, [49], which is a complexity measure specifically dealing with multivariate time series data, or other complexity measures fit to deal with multivariate problems. Further criteria for the best fit based on correlations present in multivariate data may be found regarding the fractal interpolation for multivariate time series data.

The limitations of the presented approach hide in the parameter range of the neural network implementation. Though we can set arbitrary ranges to parameterize the neural network, computation costs can be reduced significantly if a good range for a specific dataset is known or can be guessed.

Further research of the presented framework may include switching the LSTM layers to feed-forward neural network layers or simple recurrent neural network (RNN, i.e., non-LSTM layers) layers. Here, one can adopt the ideas of time-delayed recurrent neural networks, [50], or time-delayed feed-forward neural networks, [51]. For both approaches, one can choose the input of the neural network to match the embedding of the time series, i.e., use the estimated time-delay and embedding dimension as done for a phase space reconstruction of a univariate time series data (see Appendix F), as done in [52].

## 12. Conclusions

This research aimed to improve neural network time series predictions using complexity measures and interpolation techniques. We presented a new approach not existing in the recent literature and tested it on five different univariate time series.

First, we interpolated the time series data under study using fractal and linear interpolation. Second, we generated randomly parameterized LSTM (one may understand the randomly parameterized LSTM neural network as a sort of a brute-force neural network approach). Neural networks and a step-by-step approach predicted the data under study, i.e., each dataset, and consequently, a non-interpolated dataset, a linear-interpolated dataset, and a fractal-interpolated dataset. Lastly, we filtered these random ensemble predictions based on their complexity, i.e., we kept only the forecasts with complexity close to the original complexity of the data. By applying the filters to the randomly parameterized LSTM ensemble, we reduced the error of the randomly parameterized ensemble by a factor of 10. The best filtered ensemble predictions consistently outperformed a single LSTM prediction, which we use as a reasonable baseline.

Filtering ensembles based on their complexities has not been done before and should be considered for future ensemble predictions to reduce the costs of optimizing neural networks.

In terms of interpolation techniques, we found that fractal interpolation works best under the given circumstances. For the complexity filters, we identified a combination of a Single-Value-Decomposition-based, e.g., SVD entropy or Fisher’s information, and another complexity measure, e.g., the Hurst exponent, to perform best.

We conclude that interpolation techniques generating new data with a complexity close to that of the original data are best suited for improving the quality of a forecast. We expect the presented methods to be further exploited when predicting complex real-life time series data, such as environmental, agricultural, or financial data. This is because researchers are often confronted with a meager amount of data, and the given time series properties/complexity may change over time. Thus, new sets of hyper-parameters may have to be found. Using a random neural network ensemble and then filtering the predictions with respect to the complexity of older data circumvents this problem.

## Figures and Tables

**Figure 1 entropy-23-01424-f001:**
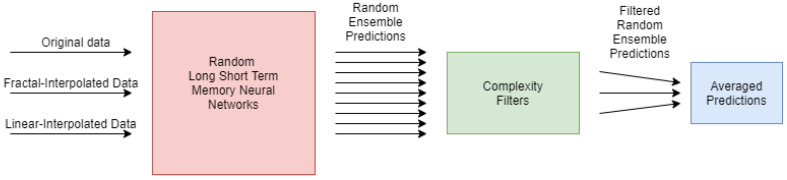
Schematic depiction of the developed pipeline. The whole pipeline is applied to three different sorts of data for each time series. First, the original non-interpolated data, second, the fractal-interpolated data, and third, the linear-interpolated.

**Figure 2 entropy-23-01424-f002:**
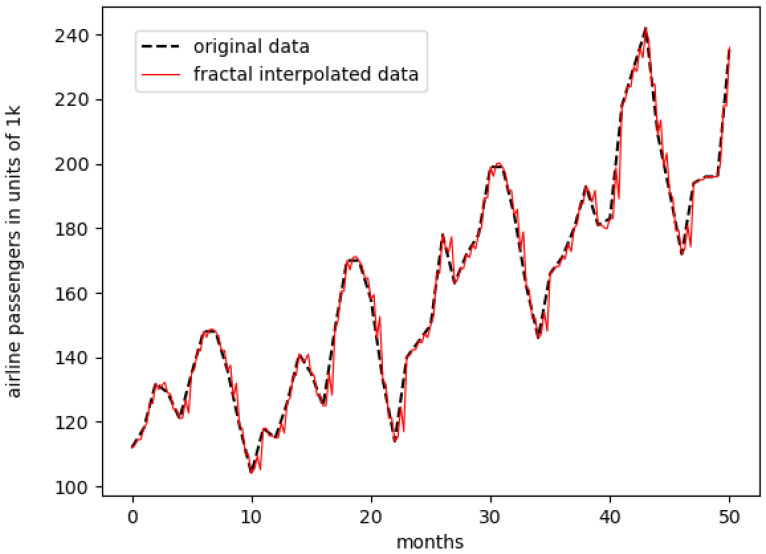
Fractal-interpolated monthly airline passengers data, first 50 data points.

**Figure 3 entropy-23-01424-f003:**
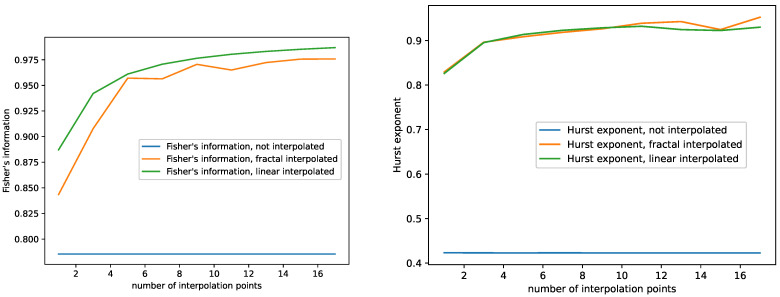
Plots for Fisher’s information and the Hurst exponent depending on the number of interpolation points for the non-interpolated, the fractal-interpolated and the linear-interpolated data. Monthly international airline passengers dataset.

**Figure 4 entropy-23-01424-f004:**
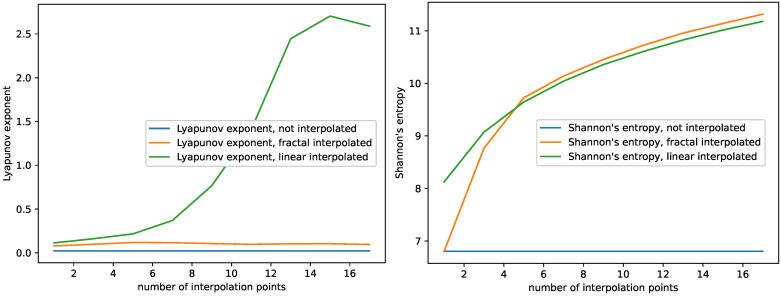
Plots for the Largest Lyapunov exponent and Shannon’s entropy depending on the number of interpolation points for the non-interpolated, the fractal-interpolated and the linear-interpolated data. Monthly international airline passengers dataset.

**Figure 5 entropy-23-01424-f005:**
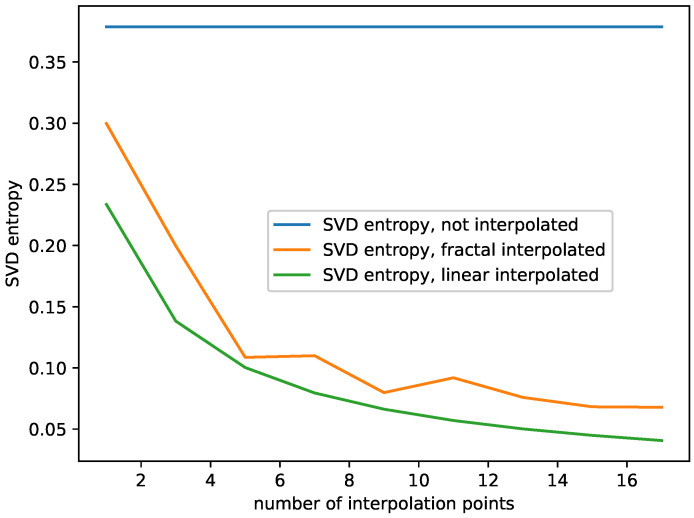
Plot for the SVD entropy depending on the number of interpolation points, for the non-interpolated, the fractal-interpolated and the linear-interpolated data. Monthly international airline passengers dataset.

**Figure 6 entropy-23-01424-f006:**
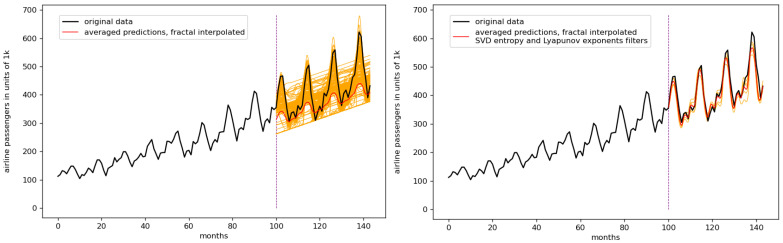
Plots for both the unfiltered ensemble predictions (left side) and the filtered ensemble prediction using a consequent application of, first, an SVD entropy filter, and second, a filter based on Lyapunov exponents to improve the prediction, 6 interpolation points. The orange lines are all the predictions constituting the ensemble, the red lines are the averaged predictions.

**Figure 7 entropy-23-01424-f007:**
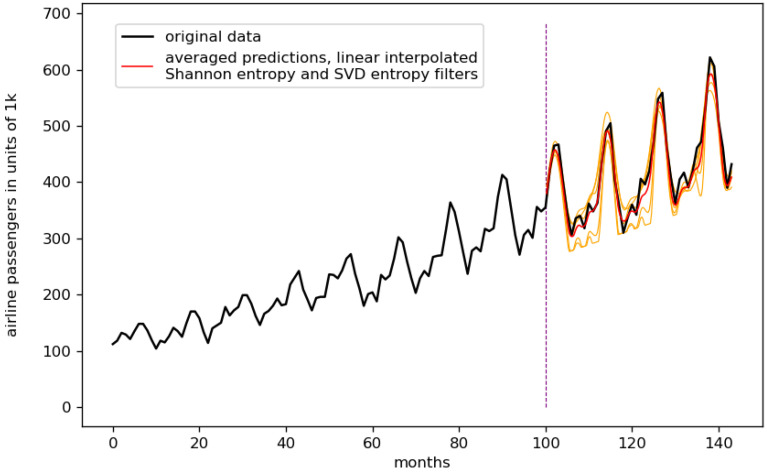
Best result monthly airline passengers dataset. The orange lines show the remaining ensemble predictions after filtering, the red line is the averaged ensemble prediction. Linear-interpolated, three interpolation points, Shannon entropy and SVD entropy filter, error: 0.03542 ± 0.00625.

**Table 1 entropy-23-01424-t001:** Complexities of the original datasets.

	Hurst Exponent	Largest Lyapunov Exponent	Fisher’s Information	SVDEntropy	Shannon’s Entropy
**Monthly international** **airline passengers**	0.4233	0.0213	0.7854	0.3788	6.8036
**Monthly car sales** **in Quebec**	0.7988	0.0329	0.5965	0.5904	6.7549
**Monthly mean** **air temperature** **in Nottingham** **Castle**	0.4676	0.0069	0.6617	0.5235	7.0606
**Perrin Freres** **monthly** **champagne sales**	0.7063	0.0125	0.3377	0.8082	6.6762
**CFE specialty** **monthly writing** **paper sales**	0.6830	0.0111	0.5723	0.6138	7.0721

**Table 2 entropy-23-01424-t002:** Baseline RMSE for all datasets, LSTM.

Dataset	Train Error	Test Error	Single Step Error
**Monthly international airline passengers**	0.04987	0.08960	0.11902
**Monthly car sales in Quebec**	0.09735	0.11494	0.12461
**Monthly mean air temperature in Nottingham Castle**	0.06874	0.06193	0.05931
**Perrin Freres monthly champagne sales**	0.07971	0.07008	0.08556
**CFE specialty monthly writing paper sales**	0.07084	0.22353	0.21495

**Table 3 entropy-23-01424-t003:** Baseline RMSE for all datasets, GRU.

Dataset	Train Error	Test Error	Single Step Error
**Monthly international airline passengers**	0.04534	0.07946	0.10356
**Monthly car sales in Quebec**	0.09930	0.11275	0.11607
**Monthly mean air temperature in Nottingham Castle**	0.07048	0.06572	0.06852
**Perrin Freres monthly champagne sales**	0.06704	0.05916	0.07136
**CFE specialty monthly writing paper sales**	0.09083	0.22973	0.23296

**Table 4 entropy-23-01424-t004:** Baseline RMSE for all datasets, RNN.

Dataset	Train Error	Test Error	Single Step Error
**Monthly international airline passengers**	0.05606	0.08672	0.10566
**Monthly car sales in Quebec**	0.10161	0.12748	0.12075
**Monthly mean air temperature in Nottingham Castle**	0.07467	0.07008	0.06588
**Perrin Freres monthly champagne sales**	0.08581	0.07362	0.07812
**CFE specialty monthly writing paper sales**	0.07195	0.22121	0.21316

**Table 5 entropy-23-01424-t005:** Error table for the monthly airline passengers dataset. The bold results are the three best ones for this dataset.

Interpolation Technique	# of Interpolation Points	Filter	Error
non-interpolated	-	fisher svd	0.04122 ± 0.00349
non-interpolated	-	svd	0.04122 ± 0.00349
non-interpolated	-	svd shannon	0.04122 ± 0.00349
non-interpolated	-	fisher	0.04166 ± 0.00271
non-interpolated	-	fisher shannon	0.04166 ± 0.00271
**fractal-interpolated**	**1**	**fisher hurst**	**0.03597 ± 0.00429**
**fractal-interpolated**	**1**	**svd hurst**	**0.03597 ± 0.00429**
fractal-interpolated	5	hurst fisher	0.03980 ± 0.00465
fractal-interpolated	5	hurst svd	0.03980 ± 0.00465
fractal-interpolated	5	shannon	0.04050 ± 0.00633
**linear-interpolated**	**3**	**shannon svd**	**0.03542 ± 0.00625**
linear-interpolated	3	shannon fisher	0.03804 ± 0.00672
linear-interpolated	5	fisher	0.04002 ± 0.00357
linear-interpolated	5	fisher shannon	0.04002 ± 0.00357
linear-interpolated	5	svd fisher	0.04002 ± 0.00357

## Data Availability

All used data sets are part of the *time series data library*, which is cited and linked in the discussion of the data sets.

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
