# Peer review of "Taming the Chaos in Neural Network Time Series Predictions"

_entropy, 2021, doi:10.3390/e23111424_

Round 1
Reviewer 1 Report
The manuscript proposed an approach to improve neural networks for time series prediction using complexity measures and interpolations. The proposed approach is interesting and results are impressive. However, it is recommended to consider the following comments to revise the manuscript: 1. All time series used in this manuscript are highly seasonal and with periodic patterns. It is advised to evaluate the performance of proposed concepts on highly chaotic time series such as wind energy/speed. 2. At the first sight, the predicted results/graphs look accurate, however, its accuracy cannot be evaluated without comparing them with the state-of-the-art forecasting methods. Therefore it is advised to compare the performance with methods such as ARIMA, RNN, PSF, etc for the better evaluation of the proposed approach. 3. In this case study, the nature of predictions such as forecast horizon, forecast strategies are not discussed clearly. Further, the comparison seems to be done on the basis of a single iteration. Such comparisons are not sufficient to decide the best method in the pool. Therefore, it is advised to such strategies such as cross-validation or monte Carlo simulations to evaluate the performance of models in a robust manner. The authors can refer to this article for more details: https://www.mdpi.com/1996-1073/13/10/2578 4. Quality of many figures are poor (especially, Figure 5). It is advised to revise them with higher resolution and better aesthetics. 5. There are several equations where all variables are not discussed in the corresponding text. It is advised to mention all details.Author Response
Please see the attachment.
Further, for the full review letter and the revised manuscript check out the uploaded full-content-file "revised-submission.zip" which was uploaded as a general response.

Reviewer 2 Report
The paper deals with a problem that is very important in machine learning (ML) – time series (TS) prediction. This problem is solved using a novel approach suggested by the authors, combining interpolation techniques, randomly parameterized LSTM neural networks (NN) and measures of signal complexity. The proposed approach is tested on five different univariate TS data. The authors prove this method to be useful for real-time TS prediction because the optimization of hyperparameters, which is usually very costly and time-intensive, can be circumvented with the presented approach. The study falls well within the scope of the Journal, and its solution will be definitely interesting for the wide audience of the Journal.
The paper consists of: an introduction, describing the importance of the problem of TS prediction, the main idea of the study and the general outline of the paper; a brief review of the related work; a short description of methodology; a list of the datasets used; a description of the applied interpolation techniques; a description of the used measures of data complexity; a description of LSTM ensemble predictions; an analysis of the used error measures; a description of the complexity filters used; a description of LSTM baseline prediction; the description and the discussion of the results; a conclusion; an acknowledgements section; five appendixes; and a reference list containing 46 references.
In the whole, the presented paper is good and is definitely publishable in Entropy. The studied problem is very topical. Use of ML approach for TS prediction is an extensive and rapidly developing field of knowledge. Although the parts of the approach used by the authors (data interpolation, random NN parameterization and TS complexity estimation) are not novel, the idea and the details of their combination look novel and are interesting enough. The presented results are original.
The paper is well written, in good English, and easy to understand. The ideas of the authors are clear. The study is technically sound. The paper has a fair balance between explanation of the known notions and their details, and referencing external sources. The ideas and the methods used are described in the paper in due detail for the reader to understand without referring the referenced sources.
However, there are several unclear items regarding the methods used that would better be addressed before the paper could be actually published.
1) All the TS used in this study are monthly with a pronounced yearly cycle. It would be very interesting to test the approach for other types of TS having no cyclicity. Will the methods of the paper work for such types of TS? Will the main conclusions of the paper hold? The list of the datasets used in the paper needs to be supplemented with non-cyclical TS.
2) All the TS used in this study are univariate. At the same time, a very common case is prediction of multivariate TS; different components of such TS may have different properties regarding also signal complexity. The authors may want to add a discussion of this case in the description of their methodology.
3) The embedding dimension for calculation of Fisher’s information criterion is set to 3 for all the TS, based on the Takens theorem. However, this does not seem justified enough. An estimation of the actual embedding dimension for each TS is possible, and it would better be performed at least for one TS to demonstrate that the rough estimation of 3 is actually reasonable.
4) LSTM paradigm is now the one used most often for TS prediction. However, there is a well-known alternative to use of recurrent NNs – use of simple feedforward NN with delay embedding of the TS. The approach used in the paper seems to be easily extendable to this case also. The authors may want to comment on this in the paper.
5) Of all the parameters of the NN used, some are randomly chosen and some are fixed. While use of rmsprop optimizer and MSE loss function seems reasonable, the used batch size of 2 looks strange enough – it may degrade the performance of the trained NN in respect to the case of not using minibatch training at all, with a very small gain in computational cost. Obviously changing this value or including the batch size in the list of random parameters would not change the results of the paper, but still it would be better for the authors to justify the choice of batch size.
6) Ref.23 should probably be complemented with an URL in order for the source to be accessible. The present reference does not allow one to uniquely identify the source.
7) In the whole, the English of the paper is good. However, there seems to be a problem with excessive or incorrect use of ‘s (other's instead of others, data's complexity instead of data complexity, it's instead of its, Taken's instead of Takens, author's instead of authors', other's instead of others etc.)
To summarize, in the opinion of the reviewer, this paper requires minor revision, after which it can be accepted for publication in Entropy.
Author Response
Please see the attachment.
Further, for the full review-letter and the revised manuscript check out the uploaded full-review-file "revised-submission.zip", which was uploaded as a general response.

Round 2
Reviewer 1 Report
Updates in the manuscript are satisfactory.